# Extracellular Matrix Remodeling in Motor Neuron Diseases

**DOI:** 10.3390/ijms262311376

**Published:** 2025-11-25

**Authors:** Savina Apolloni, Silvia Tortoriello, Martina Milani, Simona Rossi

**Affiliations:** 1Department of Biology, University of Rome Tor Vergata, 00133 Rome, Italy; silvia.tortoriello94@gmail.com (S.T.); martina.milani@uniroma2.it (M.M.); 2Institute of Translational Pharmacology (IFT), National Research Council (CNR), 00133 Rome, Italy

**Keywords:** ECM, motor neuron diseases, ALS, SMA

## Abstract

The extracellular matrix (ECM) constitutes a dynamic scaffold composed of both cellular and non-cellular elements that not only ensure tissue integrity but also regulate signaling events crucial for development and homeostasis. While its dysregulation has long been investigated in cancer, fibrosis, and autoimmunity, increasing evidence implicates ECM remodeling in neurodegenerative diseases, including motor neuron diseases (MNDs). Amyotrophic lateral sclerosis and spinal muscular atrophy, the most studied MNDs, both exhibit profound ECM alterations that influence synaptic connectivity, glial reactivity, and neuroinflammation. This review outlines recent data on ECM dynamics in MNDs, highlighting shared and disease-specific mechanisms, their potential as biomarkers, and therapeutic opportunities targeting the ECM environment to preserve neuronal function and slow disease progression.

## 1. Introduction

The extracellular matrix (ECM) is a dynamic and highly organized network made of cellular and non-cellular components, which together provide the structural and biochemical foundation for tissue development [1]. ECM molecules are synthesized in the cytoplasm and then secreted in the extracellular space, where they are further modified and differentially organized to generate tissue-specific architectures with distinct functions [2,3]. The non-cellular fraction of the ECM is mainly composed of fibrous proteins and glycoproteins. Collagens, accounting for about 90% of the ECM, provide mechanical strength and mediate signal transduction [3], while laminin, fibronectin, and elastin contribute, respectively, to cell anchoring [4,5], mechano-transduction [6], and tissue elasticity [7]. Glycoproteins, mainly represented by proteoglycans (PGs), are formed by a protein core with glycosaminoglycan (GAG) chains, and they ensure tissue hydration and resistance through their highly negative charge [3,8]. Different GAG combinations generate several PGs that regulate key cellular processes such as adhesion, migration, proliferation, and differentiation [1,9].

Beyond serving as a scaffold, the ECM acts as a signaling hub. Its composition, post-translational modifications, and mechanical properties (such as stiffness and viscoelasticity) transduce signals via integrins, CD44, and mechanoresponsive pathways (YAP/TAZ, Piezo/TRP channels), thereby regulating cell survival as well as the maintenance of tissue homeostasis and repair. Importantly, the disruption of ECM turnover can reprogram cell behavior across different tissues, underscoring the ECM as an active participant in the onset of pathological mechanisms, rather than a passive bystander [10].

To date, ECM dysregulation has been implicated in different diseases, mainly cancer, fibrosis, and autoimmune disorders [11]; however, emerging observations suggest the involvement of ECM also in neurodegenerative diseases. Indeed, the ECM shows a peculiar composition and organization in the CNS: unlike most tissues, it is characterized by a higher proportion of glycoproteins compared to fibrous proteins, with specialized hyaluronan-binding chondroitin sulphate proteoglycans (CSPGs) such as lecticans (aggrecan, neurocan, brevican, versican, and phosphacan) as major components, varying their expression across development [12]. Such molecules are organized into two main specialized ECM structures: a diffuse perisynaptic matrix that supports neural networks, and highly condensed perineuronal nets (PNNs), that surround specific neuron populations in the central nervous system (CNS) to regulate signal transduction and neuroprotection [13]. By surrounding the synapses, the ECM can modulate their puncta and receptor density at a post-synaptic level [14] as well as participate in neurotransmitters diffusion in the extracellular space [15]. PNNs, on the other hand, mainly gather around inhibitory parvalbumin-positive GABAergic inhibitory interneurons, excitatory glutamatergic neurons, pyramidal neurons in the cerebral cortex and spinal motor neurons, thus modulating neuron excitability and synaptic plasticity, axonal growth and neural regeneration [13]. Hyaluronan constitutes the backbone of PNNs, to which lecticans are condensed. The entire structure is stabilized by tenascin-R and link proteins such as HAPLN1 and HAPLN2. Due to their elevated anionic components, PNNs also play a neuroprotective role by maintaining local ion homeostasis. This is particularly relevant for parvalbumin-positive interneurons: their fast-spiking properties lead to an elevated metabolic rate, which in turn exposes them to oxidative damage [16]. The components of perisynaptic ECM are overall the same as PNNs, however tenascin and thrombospondin seem to play a more important role, as well as its fibrous fraction [13,17], sustaining the hypothesis of ECM participation in synaptogenesis [18].

Consistent with such a distinctive configuration in the CNS, ECM alterations have been observed in multiple neurodegenerative diseases. Post-mortem analyses of Alzheimer’s disease (AD) patients’ tissue show altered PG expressions [19], while fibronectin and hyaluronan are upregulated in both AD patients and mouse models. Furthermore, the expression of ECM remodeling-related enzymes improves short-term memory in mice and reduces the amount of amyloid-β deposit by enhancing astrocyte recruitment and autophagy at plaques [20]. Perisynaptic ECM was also hypothesized to play a protective role: indeed, preserving axonal coats turnover in AD synaptic degeneration may preserve synapses integrity [21]. Moreover, the analysis of Parkinson’s disease dopaminergic neurons signaling network places ECM-related pathways among the most dysregulated [22], and changes in ECM stiffness and chemical composition affect microglia morphology and survival, thereby exacerbating neuroinflammation, which contributes to the progression of the disease [23]. A similar involvement in promoting inflammation is observed in multiple sclerosis, where ECM molecules are deposited into brain lesions preventing recruitment and differentiation of oligodendrocyte progenitor cells; consistently, matricellular proteins are differentially expressed in active lesion sites [24,25].

Motor neuron diseases (MNDs) comprise a group of progressive neurological disorders characterized by the degeneration of upper motor neurons in the brain and/or lower motor neurons in the brainstem and spinal cord. The resulting loss of motor signaling leads to progressive muscle weakness and atrophy. Among MNDs, amyotrophic lateral sclerosis (ALS) and spinal muscular atrophy (SMA) are the most prevalent and best studied. Although significant advances have been made in elucidating the molecular genetics and pathophysiology of MNDs, the contribution of the ECM has received comparatively little attention. Emerging evidence indicates that ECM remodeling directly influences synaptic connectivity, glial activation, and neuroinflammation, processes that are central to motor neuron survival and function. Accordingly, the ECM not only offers opportunities for biomarker discovery but also represents a promising therapeutic target. Interventions designed to modulate ECM composition or mechanical properties may help preserve the extracellular environment in which motor neurons degenerate, stabilize neuromuscular junctions (NMJs), and ultimately slow disease progression.

## 2. ECM in Amyotrophic Lateral Sclerosis

ALS is a fatal and rapidly progressive neurodegenerative disorder characterized by the selective loss of upper and lower motor neurons. Genetic factors play a crucial role in the development of ALS. Approximately 5–10% of cases are familial (fALS), where a hereditary pattern is observed. In about 70% of these familial cases, pathogenic mutations have been identified, with *C9ORF72*, *SOD1*, *TARDBP*, and *FUS* being the most implicated genes, responsible for roughly 40%, 20%, 4%, and 3% of cases in Western populations, respectively. The majority of ALS cases (90–95%) are classified as sporadic (sALS), although rare mutations in these same genes have also been detected in some sporadic patients. Despite significant advancements in ALS research, its underlying pathogenic mechanisms remain incompletely understood, and no effective disease-modifying treatments are currently available. While traditionally viewed as a neuron-centric disease, growing evidence highlights the importance of non-cell-autonomous mechanisms involving glial cells, vascular components, and the surrounding extracellular milieu. In particular, ECM molecules have emerged as a dynamic regulator of neuronal survival, plasticity, and intercellular signaling within the CNS. Alterations in ECM composition and structure may profoundly affect neuron–glia communication, axonal maintenance, and regenerative responses, thereby contributing to disease onset and progression.

### 2.1. ECM and ALS Animal Models

In ALS rodent models, aberrant remodeling of the ECM emerges as a key pathogenic mechanism. Immunofluorescence analyses revealed abnormal expression of receptors for CSPGs, major ECM components that restrict axonal regeneration through glial scar formation. In SOD1-ALS rats, these receptors were predominantly expressed in reactive astrocytes, while neuronal expression of the CSPG receptor protein tyrosine phosphatase sigma (PTPσ) was reduced, suggesting impaired regenerative signaling between neurons and glia [26]. Consistent with these findings, proteomic and transcriptomic studies in SOD1-G93A mice revealed broad alterations in ECM-related pathways. Differentially expressed proteins and genes were enriched in ECM–receptor interactions, focal adhesion, and lysosomal function, indicating active matrix remodeling. Among these, fibronectin 1 and Fga emerged as potential biomarkers of disease onset, while the upregulation of Fmod, S100a4, S100a6, and Col1a1 further underscored the contribution of ECM dysregulation to ALS pathogenesis [27,28].

Disruption of PNNs, specialized ECM structures that protect neurons from oxidative stress, provides a mechanistic link between ECM degradation and neuronal vulnerability. In SOD1-G93A and TDP-43 Q331K mice, increased glial expression of matrix metalloprotease-9 (MMP-9) led to PNN degradation around α-motor neurons, thereby enhancing their susceptibility to oxidative damage and degeneration [29,30].

Importantly, therapeutic strategies aimed at restoring ECM integrity show encouraging results. Intraspinal transplantation of neural progenitor cells derived from induced pluripotent stem cells (NP-iPSCs) in SOD1-G93A rats at pre-symptomatic and early symptomatic stages preserved motor neurons, delayed disease progression, and extended lifespan, likely through modulation of ECM-associated genes (*versican*, *Has1*, *tenascin-R*, *Ngf*, *Igf-1*, *Bdnf*) and maintenance of PNNs [31]. Similarly, the peptide drug GM604 (Alirinetide) promoted the expression of genes involved in cell adhesion and ECM remodeling in neuronal cultures, supporting the concept that re-establishing ECM homeostasis may contribute to neuroprotection in ALS [32].

ECM integrity is also essential for maintaining the blood–spinal cord barrier (BSCB). In SOD1-G93A ALS mice, depletion of perivascular macrophages (PVMs), key components of the neurovascular unit, prevented BSCB disruption by preserving ECM protein expression required for barrier maintenance. These findings suggest that PVMs contribute to ALS pathogenesis by degrading ECM components and impairing BSCB integrity, thereby potentially accelerating motor neuron loss. Targeting PVMs to preserve ECM stability and vascular function may represent a novel therapeutic avenue for ALS [33].

### 2.2. ECM and ALS Patient-Derived Models

Recent transcriptomic studies highlight ECM remodeling as a conserved molecular feature of ALS. Analysis of gene expression profiles in motor neurons derived from C9ORF72-mutant iPSCs showed upregulation of genes involved in ECM organization, cell–matrix signaling, immune activation, and TGFβ-related pathways, mirrored at the proteomic level [34]. Similar signatures were observed in sporadic ALS, where bulk transcriptomic datasets revealed differentially expressed genes enriched in ECM structure and function. Protein–protein interaction analysis revealed tightly connected collagen-related subnetworks, suggesting that disruption of ECM integrity represents an intrinsic component of motor neuron pathology [35]. Laser capture transcriptomics (single-cell-type resolution) of post-mortem ALS spinal cords further confirmed these findings, identifying upregulated genes associated with PI3K-AKT activation, innate immune signaling, and ECM remodeling [36]. Together, these data indicate that matrix-related pathways and inflammatory signaling are co-activated during motor neuron degeneration.

Parallel bulk transcriptomic analyses of ALS astrocytes derived from human iPSCs and animal models revealed a convergent transcriptional program characterized by increased ECM remodeling and stress-response genes, accompanied by reduced expression of synaptic support and glutamate uptake machinery. This shift toward a reactive A1-like phenotype underscores the contribution of astroglial ECM dysfunction to neuronal vulnerability [37]. Multi-omics bulk studies in human prefrontal cortex and multiple ALS mouse models (C9ORF72, SOD1, TDP-43, FUS) further reinforced ECM disruption as a unifying signature of disease heterogeneity. Distinct molecular subtypes emerged, differing in immune activation, ECM dynamics, mitochondrial dysfunction, and RNA metabolism, reflecting complex but converging pathogenic mechanisms [38]. Proteomic analyses of bulk cerebrospinal fluid (CSF) from ALS patients treated with autologous mesenchymal stem cells (MSCs) revealed extensive modulation of ECM- and adhesion-related proteins post-infusion, suggesting that therapeutic benefits may involve restoration of ECM homeostasis [39].

Notably, ECM remodeling can exert dual effects. While excessive ECM deposition by reactive glia may impede repair, neuronal upregulation of ECM components could play a protective role. In C9ORF72-associated ALS/frontotemporal dementia models, dipeptide repeat expression triggered marked increases in ECM proteins such as collagen VI (COL6A1); overexpression of TGF-β1 or COL6A1 enhanced neuronal resistance to excitotoxicity, whereas their suppression exacerbated degeneration [40]. Collectively, these findings position ECM remodeling as both a driver and potential modulator of ALS pathogenesis, offering novel mechanistic and therapeutic insights.

Emerging evidence further links perivascular fibroblast activation to ECM remodeling and BSCB dysfunction in ALS. Fibroblast-derived proteins such as SPP1 and COL6A1, repeatedly identified in transcriptomic and proteomic datasets, appear to connect early vascular-matrix alterations with later neuronal degeneration. During initial disease stages, their upregulation may represent a compensatory attempt to stabilize the ECM and preserve BSCB integrity. However, chronic overproduction contributes to perivascular fibrosis, ECM stiffening, and barrier leakage, amplifying neuroinflammation and motor neuron stress. This temporal shift (from protective matrix stabilization to maladaptive fibrosis) offers a mechanistic link between ECM dysregulation, vascular impairment, and selective motor neuron vulnerability observed in both familial and sporadic ALS.

### 2.3. ECM as ALS Biomarkers

Given the strong mechanistic involvement of the ECM in ALS pathogenesis, several ECM-associated molecules have emerged as promising biomarkers for diagnosis, prognosis, and disease progression. In a large multi-cohort plasma study including 574 ALS patients across four independent populations, elevated levels of secreted phosphoprotein 1 (SPP1) at diagnosis consistently predicted shorter survival. Notably, SPP1 showed stronger prognostic value than traditional markers such as bulbar onset or CSF neurofilament levels, emphasizing its relevance to ECM dysregulation and neuroinflammatory signaling [41]. Similarly, hyaluronan, a major ECM glycosaminoglycan, correlates with disease duration and slower functional decline (ΔFRS), possibly reflecting compensatory matrix hydration and tissue remodeling [42]. CSF analyses have also identified neurocan cleavage fragments as potential diagnostic indicators [43], while matrix metalloproteinases (MMP-1/2/9), key mediators of ECM degradation and neuroinflammation, associate with accelerated neuronal loss and disease progression [44].

Expanding on these findings, a recent large-scale plasma proteomic study by Lu et al. [45] integrated Mendelian randomization with GWAS data from over 80,000 individuals (20,806 ALS cases and 59,804 controls). Nineteen plasma proteins were significantly linked to ALS risk, including ECM-related molecules such as complement component C1QC and SLITRK5, which were positively associated with disease susceptibility, whereas others, including COLEC12, showed protective associations. Functional enrichment and pathway analyses (GO and KEGG) consistently highlighted ECM organization among the major biological processes implicated in ALS, reinforcing its contribution to disease mechanisms.

Despite promising associations, reproducibility and clinical validation of ECM-related biomarkers remain limited. Most findings originate from single-cohort studies with heterogeneous methodologies. Future work should prioritize standardized quantification, longitudinal monitoring, and cross-platform validation to establish the robustness and translational utility of ECM signatures in ALS.

Collectively, these datasets support ECM remodeling as a unifying disease axis across species and modalities. Although integrative approaches combining human and animal data are necessary to address the limited overlap observed to date, comparative analyses across mouse, human, and iPSC-derived systems reveal overlapping upregulation of COL6A1, SPP1, and MMP9, suggesting that ECM remodeling is a core pathogenic feature. Notably, COL6A1 appears to be neuroprotective because its overexpression confers resistance to glutamate-induced toxicity, whereas excessive MMP activity exacerbates inflammation and barrier dysfunction. These findings highlight both detrimental and compensatory aspects of ECM remodeling in ALS.

Although ECM remodeling clearly emerges as a core feature of ALS pathology, its causal versus correlative nature remains a topic of debate. Some alterations, such as early MMP-9 activation, PNN degradation, and loss of vascular ECM components, appear to drive motor neuron degeneration actively. In contrast, others, including COL6A1 upregulation and TGF-β-related pathways, may reflect adaptive or neuroprotective responses. Clarifying these temporal and cell-specific dynamics will be crucial for distinguishing causal drivers from secondary compensatory changes and for identifying ECM-targeted interventions with genuine disease-modifying potential.

## 3. ECM in Spinal Muscular Atrophy

SMA is an autosomal recessive neurodegenerative disorder caused by loss or mutation of *SMN1* gene, leading to reduced levels of the survival motor neuron (SMN) protein and resulting in progressive lower motor neuron degeneration and muscle weakness. The loss of functional SMN protein is partially compensated by the presence of *SMN2*, a paralogous gene of *SMN1*, whose number of copies strongly influences disease severity. SMA is traditionally classified into types 0–IV based on age of onset and disease severity, ranging from severe infantile forms (type 0/I) to adult-onset, mild forms (type IV). Recent breakthroughs in gene-targeting therapies, including nusinersen, risdiplam, and onasemnogene abeparvovec, have revolutionized SMA management by restoring SMN protein production. Although these treatments have significantly improved survival and motor outcomes, particularly in early-treated patients, they remain insufficient to fully reverse disease pathology and are often less effective in older individuals, highlighting the need for novel and combinatorial strategies beyond SMN replacement.

Similarly to ALS, SMA has long been regarded as a motor neuron-autonomous disorder. However, growing evidence highlights the critical contribution of non-neuronal cells, particularly glial cells, to disease progression. ECM alterations are emerging as a critical yet underexplored aspect of SMA pathophysiology. Elucidating how these non-cell-autonomous mechanisms influence neurodegeneration may pave the way for complementary therapeutic strategies and a more comprehensive approach to SMA treatment.

### 3.1. ECM and SMA Animal Models

Microarray analysis of whole spinal cords from a severe SMA mouse model identified a broad dysregulation of genes associated with ECM integrity [46]. While only minimal transcriptional changes were detected at the pre-symptomatic stage, significant alterations became evident at the late symptomatic phase. Notably, among the 41 differentially expressed genes identified, 18 were ECM-related, including key structural and regulatory ECM components such as collagens (*Col3a1*, *Col1a1*, *Col1a2*, *Col12a1*), *laminin α2*, *fibronectin 1*, *decorin* and *periostin*, all significantly downregulated. Although it remains unclear whether these changes are causative or secondary to the disease process, they indicate that ECM integrity within the CNS is compromised in SMA and suggest that such alterations may actively contribute to disease pathogenesis. This observation was further supported by a recent systematic comparative meta-analysis integrating six independent transcriptomic studies in SMA, which identified ECM organization among the most significantly enriched gene ontology categories across multiple tissues and mouse models [47].

NMJ dysfunction represents an early hallmark of SMA pathogenesis. Recent studies in mice and zebrafish have shown that chondrolectin (Chodl), a member of the C-type lectin superfamily, directly binds the ECM component collagen XIXa1 (Col19a1), and that this interaction is essential for proper NMJ formation, as well as motor axon growth and branching [48]. Notably, Chodl is dysregulated early in SMA mouse models, prior to overt muscle weakness [49,50], and its overexpression partially rescues axonal defects in zebrafish SMA models [51]. Collectively, these findings identify the Chodl-Col19a1 interaction as a critical ECM-dependent mechanism for NMJ formation and maintenance, highlighting defects in this pathway as early contributors to SMA pathogenesis. Interestingly, aberrant Chodl expression has also been reported in a mouse ALS model [52,53], and higher expression levels of its binding partner Col19a1 correlates with faster disease progression in patients [54], underscoring the broader role of this ECM pathway across MNDs.

Importantly, ECM abnormalities are not restricted to the NMJ. Hunter et al. demonstrated that Schwann cells from SMA mouse models display intrinsic SMN-dependent defects, leading to impaired myelination, disrupted axo–glial interactions and abnormal ECM composition in peripheral nerves in vivo [55]. Moreover, Schwann cells isolated from SMA mice showed reduced expression of key myelin proteins and ECM components, including laminin α2, and failed to properly stabilize neurite extensions in co-culture with healthy motor neurons. Notably, restoration of SMN levels rescued myelin protein expression, supporting the concept that defective Schwann cell function and ECM disruption contribute to motor neuron vulnerability through non-cell-autonomous mechanisms.

Collectively, these findings provide strong evidence that ECM dysregulation across central and peripheral nervous system components contributes to SMA pathophysiology.

### 3.2. ECM and SMA Patient-Derived Models

Patient-derived SMA models offer key insights into ECM dysregulation within the CNS. Transcriptomic analyses of iPSC-derived motor neurons from SMA patients revealed consistent downregulation of genes encoding ECM components, particularly those associated with the PNN, including *tenascin C* (*TNC*) and *thrombospondin 2* (*THBS2*) [56]. Notably, the astrocyte-secreted microRNA miR-146a, known to be toxic to SMA motor neurons and upregulated in SMA iPSC-derived astrocytes [57], was able to reproduce this transcriptional signature in vitro. Functional assays demonstrated that miR-146a exposure reduces the expression of ECM-related genes, including those involved in synaptic PNN structure such as *TNC*, *THBS2*, *HAPLN1*, *NEDD9* and *ITGA4*, and is associated with alterations in spontaneous electrophysiological activity and reduced expression of synaptic-related genes [56]. These findings suggest that SMA motor neurons exhibit an altered synaptic ECM composition, and that astrocyte-secreted miR-146a may contribute to these changes at the perineuronal net, affecting synaptic stability and excitability. Consistently, HAPLN1 is significantly downregulated in patient-derived fibroblasts from severe type I compared to mild type III SMA [58]. Altogether, these data underscore ECM remodeling within the spinal cord and motor neuron microenvironment. In particular, they point to impaired PNN integrity, as a central, cell non-autonomous mechanism in SMA pathogenesis. ECM changes appear to link abnormal glial signaling to defective synaptic connectivity and increased neuronal vulnerability.

### 3.3. ECM as SMA Biomarkers

The identification of reliable biomarkers is essential for monitoring disease progression and evaluating treatment efficacy in SMA. Recent multi-omics approaches have begun to uncover distinct molecular signatures associated with the response to nusinersen, highlighting ECM-related molecules as promising candidates for biomarker development in SMA. Longitudinal CSF proteomic and metabolomic analyses from nusinersen-treated SMA type 3 patients revealed sustained molecular remodeling after treatment [59]. Using high-resolution, non-targeted mass spectrometry, 26 differentially expressed proteins were identified after 22 months of treatment, mainly related to ECM composition, cellular migration, and CNS development. Among these, the versican core protein (VCAN) showed the highest upregulation, together with other ECM-related proteins including collagen type I α2 (COL1A2). Notably, upregulation of COL1A2 levels after 10 months of nusinersen treatment was also reported in another study on SMA type 3 patients [60], and correlated with improved clinical outcome, supporting its potential as a biomarker of nusinersen efficacy [59].

Beyond proteomic alterations, extracellular RNAs and microRNAs have emerged as additional indicators of neuron and glial health following nusinersen treatment. Although overall expression levels of several microRNAs previously associated with SMA pathology improved after treatment, Welby et al. reported that astrocyte-secreted miR-146a, which modulates PNN-related gene expression, remained elevated in post-treatment CSF samples from SMA type 1/2/3 patients [56]. These findings highlight that certain glia-derived microRNAs may serve as persistent markers of astrocyte-mediated stress and ongoing synaptic vulnerability, even in treated patients. Although further validation studies are required, these findings collectively suggest that ECM components and ECM-related transcripts and microRNAs in the CSF may serve as sensitive indicators of therapeutic efficacy and cellular health, while also underscoring the contribution of astrocyte–ECM interactions to SMA pathophysiology and treatment response.

## 4. Integrative View and Mechanobiological Implications in MNDs

Although ALS and SMA differ in their genetic origins, both disorders share convergent ECM remodeling processes that deeply affect neuronal microenvironments (Figure 1).

In both conditions, PNN disruption alters neuronal excitability and plasticity, and dysregulated collagens such as COL6A1, COL1A2, and COL19A1 emerge as common denominators linking glial activation, neuroinflammation, and neuronal stress resistance. In ALS, excessive MMP activity accelerates synaptic degradation and BSCB leakage, whereas in SMA, SMN-dependent repression of ECM genes reduces synaptic anchorage and trophic support.

Beyond biochemical signaling, growing evidence from mechanobiology emphasize that ECM alterations may not only modify molecular signaling but also reshape the physical environment of motor neurons, thereby influencing excitability and survival [61,62,63]. Alterations in the mechanical properties of the ECM, including stiffness and viscoelasticity, are increasingly recognized as drivers of neuronal vulnerability in the CNS [17,61,64]. Loss of PNN compaction and changes in ECM composition modify the perineuronal mechanical niche, perturbing ion channel function and calcium dynamics and thereby promoting hyperexcitability and synaptic instability [63]. Conversely, matrix stiffening, driven by collagen deposition or glial scarring, engages mechano-sensitive pathways such as integrin/FAK and YAP/TAZ, which amplify inflammatory and degenerative signaling [61,62]. Techniques such as atomic force microscopy, rheology, and magnetic resonance elastography demonstrate how ECM rigidity and viscoelasticity influence neuronal morphology, excitability, and synaptic function, linking mechanical alterations to neurodegenerative processes [65,66]. These biophysical alterations link ECM remodeling, BSCB disruption, and progressive denervation. Although direct biomechanical measurements are still lacking in MNDs, indirect evidence including PNN disruption, chronic inflammation, and glial reactivity in ALS and SMA, suggests that changes in the local mechanical microenvironment may influence synaptic function and neuronal excitability. Collectively, these findings point to ECM mechanobiology as an underexplored contributor to motor neuron degeneration and a promising source of mechano-biomarkers and therapeutic targets in ALS and SMA [17,65]. A summary of the main ECM-related findings across ALS and SMA models and patient-derived systems is provided in Table 1.

## 5. Conclusions and Perspectives

Altogether, evidence from ALS and SMA models converges on a maladaptive ECM phenotype that may promote both inflammation and mechanical stiffening of the neural niche. This shared pathological pattern highlights ECM remodeling as a central hub in the progression of MNDs, rather than a secondary process. Recent genetic findings further reinforce this concept. Pathogenic variants in the *von Willebrand factor A domain containing 1* (*VWA1*) gene, which encodes an ECM protein localized in basement membranes particularly in muscle and nervous system, have been linked to a hereditary motor neuropathy showing partial overlap with motor neuron disease [67,68,69]. Functional studies in zebrafish and murine models show that loss of VWA1 affects motor neuron axonal growth, NMJ formation, and locomotor behavior, indicating a direct role for this ECM-associated protein in motor neuron maintenance [67,70].

Recognizing ECM remodeling as a unifying mechanism in MNDs opens new therapeutic perspectives. Potential interventions include enzymatic modulation, through controlled use of chondroitinase ABC or selective MMP inhibitors, to rebalance proteolytic activity [61]. Matrix restoration strategies leveraging mesenchymal or neural stem-cell-derived ECM factors could reinforce PNNs and stabilize the BSCB [71]. Targeting the integrin/fibronectin axis may modulate adhesion signaling and glial mechano-sensing. Mechanical modulation using biomaterial scaffolds with physiological elasticity could normalize ECM stiffness and dampen astrocytic hyperactivation [65]. Together, these approaches aim to restore both structural and signaling balance within the neurovascular ECM. Translating ECM-targeting therapies to clinical practice requires caution. Over-inhibition of proteolytic enzymes may hinder necessary remodeling during repair. Excessive matrix deposition could impair diffusion and regeneration. Hybrid approaches that combine anti-inflammatory control, ECM modulation, and neurotrophic support may provide optimal outcomes.

Beyond therapy, ECM-related molecules emerge as biomarkers for disease monitoring and treatment response. Plasma SPP1 levels robustly predict survival in ALS. VCAN and COL1A2 correlate with response to nusinersen in SMA, linking ECM composition to clinical outcome. Integrating these molecular readouts with imaging markers of tissue stiffness or neuroinflammation could yield multidimensional disease signatures. This approach supports precision medicine strategies.

From a translational perspective, future research should bridge the molecular and biomechanical domains. Combining single-cell multi-omics, atomic force microscopy, and spatial transcriptomics will enable the mapping of ECM remodeling at subcellular resolution and the identification of cell-type-specific contributors. Understanding how ECM mechano-transduction pathways cross-regulate inflammation and excitotoxicity could reveal critical windows for intervention.

In conclusion, the ECM in ALS and SMA is not a passive scaffold. It actively determines neurodegeneration by integrating biochemical and mechanical cues that govern neuronal fate. Therapeutic rebalancing of ECM composition and function holds promise for restoring homeostasis in diseased spinal cord and motor circuits. Considering ECM remodeling as both a biomarker and a modifiable pathological driver will help shape future research. This perspective can support the development of ECM-based neuroprotective strategies that may change the course of MNDs.

## Figures and Tables

**Figure 1 ijms-26-11376-f001:**
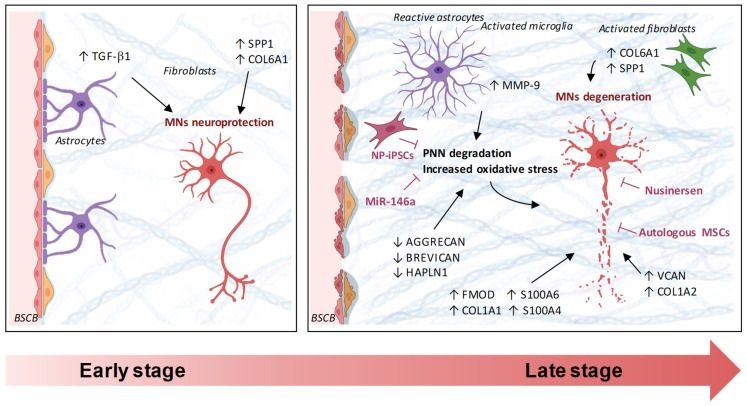
Proposed mechanism of pathological ECM dysfunction in ALS and SMA. During early stages, ECM remodeling acts as an adaptive and neuroprotective process. Increased TGF-β1 signaling and early fibroblast-derived ECM changes (e.g., elevated SPP1 and COL6A1) help stabilize the extracellular matrix, support synaptic maintenance, preserve BSCB integrity, and promote motor neuron survival. As the disease progresses, a clear transition emerges between these adaptive ECM responses and maladaptive remodeling processes. In the late stage, persistent upregulation of ECM-associated molecules (e.g., SPP1, COL6A1, COL1A2, VCAN, MMP-9), together with glial activation, including reactive astrocytes, activated microglia, and fibroblasts, drives pathological matrix reorganization. Excessive protease activity, PNN degradation, oxidative stress, and progressive ECM stiffening contribute to fibrotic deposition and loss of matrix homeostasis. Concomitant BSCB disruption further amplifies neuroinflammatory signaling, ultimately resulting in motor neuron degeneration. Therapeutic interventions such as NP-iPSCs, autologous MSCs, and nusinersen can partially restore ECM homeostasis and support motor neuron stability.

**Table 1 ijms-26-11376-t001:** ECM alterations across ALS and SMA models.

Disease	Model/Source	ECM Molecules	Function	Refs.
ALS	SOD1-G93A mouse	Fn1, Fga, Col1a1, Fmod	Structural/Adhesion	[27,28]
ALS	TDP-43 Q331K mouse	MMP-9 ↑; PNN degradation	Synaptic ECM	[29,30]
ALS	C9ORF72 KI mouse	Col6a1 ↑	Structural/Protective	[40]
ALS	iPSC motor neurons (C9, sALS)	COL6A1, SPP1	Structural/Signaling	[34,35]
ALS	LCM MNs (post-mortem)	ECM remodeling genes	Structural/Signaling	[36]
ALS	Patient CSF (MSC-treated)	APOA1, APP, C4A, FGA, FGG	Repair pathways	[39]
ALS	Plasma/serum	SPP1, hyaluronan	Biomarkers	[41,42]
SMA	Severe mouse	ECM-related genes	Structural/Signaling	[46,47]
SMA	NMJ (zebrafish/mouse)	Chodl-Col19a1 axis	Synaptic ECM	[48,49,50,51]
SMA	iPSC motor neurons	TNC ↓; THBS2 ↓	Synaptic ECM	[56]
SMA	Patient fibroblasts/CSF	HAPLN1 ↓; VCAN ↑; COL1A2 ↑	Structural/Signaling	[58,59,60]

Functional groups were assigned based on the predominant biological roles reported for each ECM molecule in the cited studies. ‘Structural/Adhesion’ molecules maintain tissue architecture and mediate cell–matrix interactions; ‘Structural/Protective’ molecules provide structural support and neuroprotective functions; ‘Structural/Signaling’ molecules contribute to tissue structure and participate in cell signaling pathways; ‘Synaptic ECM’ includes proteins involved in synapse formation, maintenance, or remodeling; ‘Repair pathways’ participate in tissue repair and regeneration; and ‘Biomarkers’ refers to ECM components detected in biofluids that may serve as disease indicators. ↑: upregulated; ↓: downregulated; LCM MNs: Laser capture micro-dissected motor neurons; CSF: cerebrospinal fluid; NMJ: neuromuscular junction.

## Data Availability

No new data were created or analyzed in this study. Data sharing is not applicable to this article.

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
