# Peer review of "Extracellular Matrix Remodeling in Motor Neuron Diseases"

_ijms, 2025, doi:10.3390/ijms262311376_

Round 1

Reviewer 1 Report

Comments and Suggestions for Authors

The review is devoted to the assertion that in motor neuron diseases, remodeling of the extracellular matrix occurs. In the abstract, the authors declare that the review will present data into ECM dynamics in MNDs, and will emphases their potential as sources of biomarkers and as a promising target for therapeutic intervention in these diseases.

The authors provide examples of disparate changes in various ECM markers and the expression of a wide range of genes. The examples represent a series of lists of changes in various indicators in neuromuscular diseases that are little related to each other.

The review consists of a number of chapters, such as: 2.1. ECM and ALS animal models,  2.2. ECM and ALS patient models, 2.2. ECM as ALS biomarkers. The chapters appear disjointed, lacking a unifying theme, with no conclusion at the end or connections between chapters. The parameters and markers discussed in animals and humans appear disjointed. In most cases, it's unclear how they can be used in therapeutic practice.

In the picture (Figure 1. Aberrant ECM remodeling in ALS) it is also worth noting that human and animal studies have few overlapping markers and components. Given that there is no general conclusion in the text, it appears as disparate studies, loosely related to each other and placed next to each other in the order in which they are listed.

The chapter 3.1. ECM and  SMA models not divided into the same sections (for animals and humans), but also lack sufficient generalizations.

The review barely touches on the mechanisms that offer insights into the action of various ECM-related molecule markers for the treatment of these diseases.

Author Response

We sincerely thank the Reviewer for the careful and constructive evaluation of our manuscript. Below, we provide detailed, point-by-point responses and highlight the changes implemented in the revised version (shown in red).

The review is devoted to the assertion that in motor neuron diseases, remodeling of the extracellular matrix occurs. In the abstract, the authors declare that the review will present data into ECM dynamics in MNDs, and will emphases their potential as sources of biomarkers and as a promising target for therapeutic intervention in these diseases. The authors provide examples of disparate changes in various ECM markers and the expression of a wide range of genes. The examples represent a series of lists of changes in various indicators in neuromuscular diseases that are little related to each other.

The review consists of a number of chapters, such as: 2.1. ECM and ALS animal models,  2.2. ECM and ALS patient models, 2.2. ECM as ALS biomarkers. The chapters appear disjointed, lacking a unifying theme, with no conclusion at the end or connections between chapters. The parameters and markers discussed in animals and humans appear disjointed. In most cases, it's unclear how they can be used in therapeutic practice.

We thank the Reviewer for this valuable observation. We have substantially revised the whole structure to provide stronger conceptual integration throughout the manuscript. Specifically, we added subsection in 2.2 lines 191-200; in 2.3 lines 225-246; new paragraph 4 “Integrative view and mechanobiological implications in MNDs”; new Figure 1 and Table 1; expanded the final paragraph 5 “Conclusions and perspectives” to highlight translational implications and therapeutic potential.

In the picture (Figure 1. Aberrant ECM remodeling in ALS) it is also worth noting that human and animal studies have few overlapping markers and components. Given that there is no general conclusion in the text, it appears as disparate studies, loosely related to each other and placed next to each other in the order in which they are listed.

We appreciate this insightful remark. We have now added a discussion on this point on page 230-238 and revised the entire Figure 1 to highlight mechanistic pathways.

The chapter 3.1. ECM and  SMA models not divided into the same sections (for animals and humans), but also lack sufficient generalizations.

We thank the Reviewer for pointing this out. Section 3 has been reorganized to parallel the ALS section, now divided into (i) ECM and SMA animal models; (ii) ECM and SMA patient-derived models and (iii)ECM as SMA biomarkers, ensuring structural coherence. Furthermore, we added a comparative paragraph across both ALS and SMA (354-371).

The review barely touches on the mechanisms that offer insights into the action of various ECM-related molecule markers for the treatment of these diseases.

We fully agree and have now strengthened this aspect throughout the manuscript. Specifically, we have incorporated new mechanistic perspectives across several sections and added some paragraphs (lines 191–200; 225–246; 354–371). We have also revised Figure 1 to better illustrate these mechanistic links. In addition, we expanded the therapeutic implications in the Conclusion (lines 409–422).

Reviewer 2 Report

Comments and Suggestions for Authors

This review article offers an extensive synthesis of known data on extracellular matrix remodeling within motor neuron diseases, particularly in the context of amyotrophic lateral sclerosis and spinal muscular atrophy. The article appears very well organized and offers an update on the current knowledge on those processes combining data from the transcriptome and cell levels. The subject matter of the paper appears particularly relevant since there has been an increasing appreciation of the role of the extracellular matrix within neurodegeneration. However, while the review is generally clear and informative, there are several issues related to depth of mechanistic discussion, integration of concepts, and figure/table presentation that limit its impact. Addressing these would significantly strengthen the manuscript’s clarity and scholarly value.

Major Comments

1. The introduction sets the stage on the relevance of ECM in general biological research very effectively. The shift from understanding the relevance of having an understanding of ECM in usual biological fields to understanding the relevance of ECM remodeling in understanding MNDs a bit more can be more compact.

2. The manuscript summarizes numerous studies but rarely provides mechanistic synthesis. For example, how ECM changes (e.g., stiffness, proteoglycan sulfation) mechanistically influence motor neuron vulnerability or glia–neuron crosstalk remains underexplored. A schematic summarizing these pathways would help.

3. Much of the content is descriptive. The authors should offer more critical interpretation, e.g., which ECM alterations are causal versus correlative? Are there contradictory findings across ALS and SMA models?

4. Interestingly, a high degree of compartmentalization also exists between animal models, human data, and in vitro observations. The paper could profit from an analysis assaying the similarities and discrepancies within the three domains and highlighting the shared ECM pathways (e.g. COL6A1 and SPP1).

5. The conclusion briefly discussed targets such as ECM but did not explore in depth the potential of using it or anything that could modulate it like enzyme inhibitors and stem cell-derived factors of the extracellular matrix. This should be expanded.

6. Table 1 is useful but overly dense. Consider separating ALS and SMA sections with clearer formatting or adding functional categorization (e.g., structural, enzymatic, signaling ECM components). Figure 1 lacks mechanistic labeling and visual clarity, please enhance legends and graphical resolution.

7. Although citations are very extensive, including more contemporary scholarship on ECM within CNS injury and neuroregeneration (for example, perineuronal net remodeling or ECM mechanotransduction within glia) would increase contextual breadth.

8. The ALS section is comprehensive but somewhat repetitive. Some paragraphs (e.g., lines 140-220) reiterate similar findings about ECM gene enrichment. Condense overlapping content and instead highlight the novel conclusions emerging from meta-analyses.

9. The manuscripts could have attempted to compare the changes within the extracellular matrix of both ALS and SMA. What conserved and disease-specific pathways exist? The synthesis could increase the level of originality.

10. Ensure consistent use of gene/protein symbols (e.g., COL6A1 vs. Col6a1). Italicize gene symbols and standardize species notation (human vs. mouse).

11. When describing multi-omics studies, specify whether findings derive from bulk tissue or single-cell datasets. This context affects the interpretation of ECM-related gene enrichment.

12. The biomarker component (SPP1, hyaluronan, neurocan) is very well-written but could be improved by considering the validation status. Are these findings reproducible in an independent set of subjects or more exploratory?

Comments on the Quality of English Language

Overall English is fluent, but several sentences are overly long or contain redundant qualifiers (“extremely peculiar,” “highly organized network”). Moderate the tone and simplify phrasing to improve readability.

Author Response

We sincerely thank the Reviewer for the careful and constructive evaluation of our manuscript. Below, we provide detailed, point-by-point responses and highlight the changes implemented in the revised version (shown in red).

This review article offers an extensive synthesis of known data on extracellular matrix remodeling within motor neuron diseases, particularly in the context of amyotrophic lateral sclerosis and spinal muscular atrophy. The article appears very well organized and offers an update on the current knowledge on those processes combining data from the transcriptome and cell levels. The subject matter of the paper appears particularly relevant since there has been an increasing appreciation of the role of the extracellular matrix within neurodegeneration. However, while the review is generally clear and informative, there are several issues related to depth of mechanistic discussion, integration of concepts, and figure/table presentation that limit its impact. Addressing these would significantly strengthen the manuscript’s clarity and scholarly value.

Major Comments

  1. The introduction sets the stage on the relevance of ECM in general biological research very effectively. The shift from understanding the relevance of having an understanding of ECM in usual biological fields to understanding the relevance of ECM remodeling in understanding MNDs a bit more can be more compact.

We thank the Reviewer for this suggestion. The Introduction has been condensed to streamline the transition from general ECM biology to its relevance in MNDs.

  1. The manuscript summarizes numerous studies but rarely provides mechanistic synthesis. For example, how ECM changes (e.g., stiffness, proteoglycan sulfation) mechanistically influence motor neuron vulnerability or glia–neuron crosstalk remains underexplored. A schematic summarizing these pathways would help.

We fully agree and have now integrated this aspect throughout the whole manuscript. Specifically, we have incorporated new mechanistic perspectives across several sections and added some paragraphs (lines 191–200; 225–246; 354–371). We have also revised Figure 1 to better illustrate these mechanistic links.

  1. Much of the content is descriptive. The authors should offer more critical interpretation, e.g., which ECM alterations are causal versus correlative? Are there contradictory findings across ALS and SMA models?

We appreciate these important points. We discussed them on lines 239-246; 275-277; 398-401 and discussed an integrative view across ALS and SMA in a new paragraph 4.

  1. Interestingly, a high degree of compartmentalization also exists between animal models, human data, and in vitro observations. The paper could profit from an analysis assaying the similarities and discrepancies within the three domains and highlighting the shared ECM pathways (e.g. COL6A1 and SPP1).

We thank the Reviewer for highlighting this point. We have added a comparative subsection on lines 230-238.

  1. The conclusion briefly discussed targets such as ECM but did not explore in depth the potential of using it or anything that could modulate it like enzyme inhibitors and stem cell-derived factors of the extracellular matrix. This should be expanded.

We fully agree. We have now expanded the final section by creating a new subsection within the Conclusions (409-442).

  1. Table 1 is useful but overly dense. Consider separating ALS and SMA sections with clearer formatting or adding functional categorization (e.g., structural, enzymatic, signaling ECM components). Figure 1 lacks mechanistic labeling and visual clarity, please enhance legends and graphical resolution.

We thank the Reviewer for this useful comment. We have revised Table 1 into two sections, ALS and SMA, with functional categorization. Figure 1 has been completely revised to include mechanistic labels.

  1. Although citations are very extensive, including more contemporary scholarship on ECM within CNS injury and neuroregeneration (for example, perineuronal net remodeling or ECM mechanotransduction within glia) would increase contextual breadth.

We agree and we have now added and discussed new recent references (e.g., Rocha et al. 2022; Di et al., 2023 Tewari et al., 2024; Ortega et al., 2024; Pillai et al., 2024), on lines 372-390 in the new paragraph 4 “Integrative view and mechanobiological implications in MNDs”.

  1. The ALS section is comprehensive but somewhat repetitive. Some paragraphs (e.g., lines 140-220) reiterate similar findings about ECM gene enrichment. Condense overlapping content and instead highlight the novel conclusions emerging from meta-analyses.

We have shortened overlapping descriptions, and We have now added the conclusion and perspectives on lines 225-246.

  1. The manuscripts could have attempted to compare the changes within the extracellular matrix of both ALS and SMA. What conserved and disease-specific pathways exist? The synthesis could increase the level of originality.

We have addressed this point by adding a comparative paragraph 4 “Integrative view and mechanobiological implications in MNDs”.

  1. Ensure consistent use of gene/protein symbols (e.g., COL6A1 vs. Col6a1). Italicize gene symbols and standardize species notation (human vs. mouse).

We carefully reviewed the entire manuscript and standardized all gene and protein nomenclature.

  1. When describing multi-omics studies, specify whether findings derive from bulk tissue or single-cell datasets. This context affects the interpretation of ECM-related gene enrichment.

We have now specified the experimental context for each multi-omics study (bulk vs. single-cell vs. laser microdissection) to enhance interpretability.

  1. The biomarker component (SPP1, hyaluronan, neurocan) is very well-written but could be improved by considering the validation status. Are these findings reproducible in an independent set of subjects or more exploratory?

We appreciate this suggestion. The section has been expanded (lines 225-229).

Overall English is fluent, but several sentences are overly long or contain redundant qualifiers (“extremely peculiar,” “highly organized network”). Moderate the tone and simplify phrasing to improve readability.

We have revised the text for conciseness, shortening overly complex sentences and reducing redundancy. The entire manuscript has undergone thorough language editing for clarity and readability.

Reviewer 3 Report

Comments and Suggestions for Authors

This review surveys how extracellular matrix (ECM) remodeling contributes to the pathogenesis of motor neuron diseases (MNDs), with emphasis on amyotrophic lateral sclerosis (ALS) and spinal muscular atrophy (SMA). The manuscript outlines basic ECM biology, then synthesizes clinical and preclinical data on ECM changes, candidate biomarkers, and therapeutic angles, closing with high-level conclusions about ECM as more than a bystander. The scope is timely and clinically relevant. 

  1. The manuscript mentions ECM composition but underplays measurable changes in stiffness/viscoelasticity, PNN compaction, and their links to hyperexcitability/denervation. A concise mechanobiology box (assays, reported magnitude of change, functional readouts) would materially strengthen the review.
  2. The ALS section would benefit from a clearer through-line—from perivascular fibroblast signatures (SPP1, COL6A1) to ECM protein changes, to BSCB disruption and motor neuron vulnerability—ideally with early vs. late disease staging.   
  3. SMA coverage leans peripheral/NMJ; please expand central ECM/PNN changes (aggrecan, brevican, tenascins, HAPLN1) and their synaptic consequences in spinal cord, mirroring the ALS depth. The miR-146a/PNN-gene paragraph is an excellent anchor—build it out with core PNN components and functional data. 
  4. Aggregate ECM-targeting ideas into a discrete subsection: enzymatic modulators (e.g., chondroitinase), MMP targeting, hyaluronidases, integrin/fibronectin axes, cell-based matrix restoration, and risks (barrier compromise, off-tissue remodeling). Right now the concepts are scattered; a consolidated roadmap would be highly useful. 
  5. Ensure consistent use of “hyaluronan” (vs. “hyaluronic acid” if both appear), define all abbreviations at first mention, and align with a final abbreviation list. (Your ALS biomarker paragraph consistently uses “hyaluronan,” which is good—keep it uniform.).
  6. Trim general ECM biochemistry and refocus on neuro-specific ECM (PNNs, perisynaptic matrix) to make room for mechanobiology and therapy sections. 
  7. Avoid repeating the same biomarker findings in text and again verbatim in the table; instead, let the table carry detail while the text interprets.

Author Response

We sincerely thank the Reviewer for the careful and constructive evaluation of our manuscript. Below, we provide detailed, point-by-point responses and highlight the changes implemented in the revised version (shown in red).

This review surveys how extracellular matrix (ECM) remodeling contributes to the pathogenesis of motor neuron diseases (MNDs), with emphasis on amyotrophic lateral sclerosis (ALS) and spinal muscular atrophy (SMA). The manuscript outlines basic ECM biology, then synthesizes clinical and preclinical data on ECM changes, candidate biomarkers, and therapeutic angles, closing with high-level conclusions about ECM as more than a bystander. The scope is timely and clinically relevant.

1. The manuscript mentions ECM composition but underplays measurable changes in stiffness/viscoelasticity, PNN compaction, and their links to hyperexcitability/denervation. A concise mechanobiology box (assays, reported magnitude of change, functional readouts) would materially strengthen the review.

We fully agree. We have now incorporated a dedicated paragraph 4 titled “Integrative view and mechanobiological implications”

2. The ALS section would benefit from a clearer through-line—from perivascular fibroblast signatures (SPP1, COL6A1) to ECM protein changes, to BSCB disruption and motor neuron vulnerability—ideally with early vs. late disease staging.

We appreciate this excellent observation. We have reorganized the entire text to emphasize the temporal sequence of ECM alterations (see for instance 191-200; 239-246; 366-371) and added the revised Figure 1 highlighting temporal changes.

3. SMA coverage leans peripheral/NMJ; please expand central ECM/PNN changes (aggrecan, brevican, tenascins, HAPLN1) and their synaptic consequences in spinal cord, mirroring the ALS depth. The miR-146a/PNN-gene paragraph is an excellent anchor—build it out with core PNN components and functional data.

We thank the Reviewer for pointing this out. We have significantly expanded the SMA section to include central nervous system ECM and PNN alterations (269-277; 306-324).

4. Aggregate ECM-targeting ideas into a discrete subsection: enzymatic modulators (e.g., chondroitinase), MMP targeting, hyaluronidases, integrin/fibronectin axes, cell-based matrix restoration, and risks (barrier compromise, off-tissue remodeling). Right now the concepts are scattered; a consolidated roadmap would be highly useful.

We appreciate this suggestion. We have now added a new subsection within the Conclusions and perspectives (409-422).

5. Ensure consistent use of “hyaluronan” (vs. “hyaluronic acid” if both appear), define all abbreviations at first mention, and align with a final abbreviation list. (Your ALS biomarker paragraph consistently uses “hyaluronan,” which is good—keep it uniform.).

We have standardized terminology throughout the manuscript. All abbreviations are now defined at first mention and listed alphabetically in the final Abbreviation section.

6. Trim general ECM biochemistry and refocus on neuro-specific ECM (PNNs, perisynaptic matrix) to make room for mechanobiology and therapy sections.

We agree and have substantially reduced the general ECM biochemistry content, refocusing the section on neuro-specific ECM components. This allowed us to expand the mechanobiology (new Paragraph 4) and therapeutic perspectives accordingly (lines 409-442).

7. Avoid repeating the same biomarker findings in text and again verbatim in the table; instead, let the table carry detail while the text interprets.

We have revised Table 1 and the entire manuscript to eliminate redundancy.

Round 2

Reviewer 1 Report

Comments and Suggestions for Authors

The review has been significantly revised and improved. The questions raised have been answered.

Author Response

The review has been significantly revised and improved. The questions raised have been answered.

We thank the Reviewer for the positive feedback.

Reviewer 2 Report

Comments and Suggestions for Authors

The authors have managed to respond well to all the concerns, big and small, raised in the last round of reviews. The current version of the manuscript has improved immensely in terms of clarity, organization, and scientific content. Among the greatest improvements in this revised version are the addition of mechanistic details in various sections, condensation of the Introduction, better coordination between ALS and SMA comparative analyses, and a dramatic improvement in the quality of Figure 1 and Table 1. The text has also improved in terms of therapeutic implications, validity discussion, and use of proper naming conventions. The current manuscript offers a more unified, conceptually rich, and enlightened discussion of the issue of extracellular matrix remodeling in light of MND. In total, the changes made enhance the quality of the article, and the text is publishable pending the following minor issues.

Minor Comment

1. A few paragraphs might also profit from slightly shorter sentence structures, especially in the sections of the conclusion where several thoughts are merged in single sentences.

2. While Figure 1 has improved, the legend could use a touch more explanation as to the difference between the depiction of maladaptive versus adaptive ECM in the figure.

3. The table, Table 1, appears much easier to understand, but it would help if a brief explanatory note were added as to how the functional groups were determined.

4. In the mechanobiology portion, an example of the measurement of stiffness or viscoelasticity in related applications in the CNS could help explain the process for those not well-versed in it.

5. Please cite the article Multipotent Mesenchymal Stem Cell-Based Therapies for Spinal Cord Injury: Current Progress and Future Prospects (Zeng et al., Biology 2023) in an appropriate context. I recommend placing it in the section discussing ECM-modulating therapies in the Conclusions (lines 409-442), specifically where you discuss stem-cell derived extracellular matrix factors and biomaterials that alter ECM mechanics.

Author Response

The authors have managed to respond well to all the concerns, big and small, raised in the last round of reviews. The current version of the manuscript has improved immensely in terms of clarity, organization, and scientific content. Among the greatest improvements in this revised version are the addition of mechanistic details in various sections, condensation of the Introduction, better coordination between ALS and SMA comparative analyses, and a dramatic improvement in the quality of Figure 1 and Table 1. The text has also improved in terms of therapeutic implications, validity discussion, and use of proper naming conventions. The current manuscript offers a more unified, conceptually rich, and enlightened discussion of the issue of extracellular matrix remodeling in light of MND. In total, the changes made enhance the quality of the article, and the text is publishable pending the following minor issues.

We thank the Reviewer for the constructive comments on our manuscript. Below, we provide detailed, point-by-point responses and indicate the changes made in the revised version (highlighted in yellow).

Minor Comment

1. A few paragraphs might also profit from slightly shorter sentence structures, especially in the sections of the conclusion where several thoughts are merged in single sentences.

We agree with the reviewer. In the revised manuscript, we have edited several sentences in the Conclusions, as well as in the main text, to improve clarity and readability (lines 65-66; 323-326; 431-464).

2. While Figure 1 has improved, the legend could use a touch more explanation as to the difference between the depiction of maladaptive versus adaptive ECM in the figure.

We thank the reviewer for this helpful suggestion. We have expanded the legend of Figure 1 as suggested.

3. The table, Table 1, appears much easier to understand, but it would help if a brief explanatory note were added as to how the functional groups were determined.

As suggested, we have added a short explanatory note below Table 1.

4. In the mechanobiology portion, an example of the measurement of stiffness or viscoelasticity in related applications in the CNS could help explain the process for those not well-versed in it.

We thank the reviewer for this recommendation. We have now added this aspect in lines 393-396.

5. Please cite the article Multipotent Mesenchymal Stem Cell-Based Therapies for Spinal Cord Injury: Current Progress and Future Prospects (Zeng et al., Biology 2023) in an appropriate context. I recommend placing it in the section discussing ECM-modulating therapies in the Conclusions (lines 409-442), specifically where you discuss stem-cell derived extracellular matrix factors and biomaterials that alter ECM mechanics.

We appreciate the reviewer’s suggestion. The recommended citation has been added to the Conclusions section, along with other relevant references addressing ECM-modulating therapies.

Reviewer 3 Report

Comments and Suggestions for Authors

The authors have addressed all the concerns raised in the previous version of the manuscript. In its current form the article now meets all the publication criteria for IJMS

Author Response

The authors have addressed all the concerns raised in the previous version of the manuscript. In its current form the article now meets all the publication criteria for IJMS

We thank the Reviewer for the positive feedback.